

# Comparison of automatic airway analysis function of Invivo5 and Romexis software

Noorshaida Kamaruddin[1], Firdaus Daud[1], Asilah Yusof[1], Mohd Ezane Aziz[2] and Zainul A. Rajion[1,3]

[1] School of Dental Sciences, Universiti Sains Malaysia, Health Campus, Kota Bharu, Kelantan, Malaysia
[2] School of Medical Sciences, Universiti Sains Malaysia Health Campus, Kota Bharu, Kelantan, Malaysia
[3] College of Dentistry, King Saud University, Riyadh, Saudi Arabia

## ABSTRACT

**Background**. Visualization and calculation of the airway dimensions are important because an increase of airway resistance may lead to life-threatening emergencies. The visualization and calculation of the airway are possible using radiography technique with their advance software. The aim of this study was to compare and to test the reliability of the measurement of the upper airway volume and minimum area using airway analysis function in two software.

**Methods**. The sample consisted of 11 cone-beam computed tomography (CBCT) scans data, evaluated using the Invivo5 (Anatomage) and Romexis (version 3.8.2.R, Planmeca) software which afford image reconstruction, and airway analysis. The measurements were done twice with one week gap between the two measurements. The measurement obtained was analyzed with $t$-tests and intraclass correlation coefficient (ICC), with confidence intervals (CI) was set at 95%.

**Results**. From the analysis, the mean reading of volume and minimum area is not significantly different between Invivo5 and Romexis. Excellent intrarater reliability values were found for the both measurement on both software, with ICC values ranging from 0.940 to 0.998.

**Discussion**. The results suggested that both software can be used in further studies to investigate upper airway, thereby contributing to the diagnosis of upper airway obstructions.

Corresponding author
Zainul A. Rajion, zar5057@gmail.com

## INTRODUCTION

The airway is a system that consists of tubes that convey inhaled air from the nose and mouth into the lungs. The skeletal support for the airway is superiorly provided by the cranial base, posteriorly provided by the spine, anterosuperiorly provided by nasal septum, and anteriorly provided by the jaws and hyoid bone. An obstruction of the upper airway will increase airway resistance and can be minor or life-threatening emergencies which require immediate medical attention. Due to this reason, airway obstructions require attention. Therefore, visualization and calculation of the airway dimensions are important. Airway
obstruction is not diagnosed with imaging; however, imaging plays a role in the anatomic assessment of the airway and adjacent structures as imaging can identify the patients with airways who are at risk for obstruction. The upper airway can be visualized on conventional computed tomography (CT), cone beam CT (CBCT) and magnetic resonance imaging (MRI).

## CBCT and image analysis

CBCT systems have been developed specifically for the maxillofacial region with the advantage of the reduced radiation doses compared with conventional CT (*Ghoneima & Kula, 2013*). Accurate and easy evaluation of the airway anatomy has been possible using those CBCT systems (*El & Palomo, 2010*). There were many studies (*Feng et al., 2015*; *Glupker et al., 2015*; *Iwasaki et al., 2009*; *Kim et al., 2010*; *Camacho, Capasso & Schendel, 2014*; *Zinsly et al., 2010*; *Ogawa et al., 2005*) of the upper airway were analyzed or assessed using CBCT. The next level up of CBCT is the advanced software tools involve airway tracing features that give the user the capability to delineate the airway's boundaries, measure its volume, and calculate and locate the minimum area (*Chenin, 2015*).

Although numerous methods with 2-dimensional (2D) cephalograms, providing limited data such as linear and angular has been proposed for upper airway studies, there were studies that evaluate the airway have introduced the use of CBCT, which made the 3D diagnosis of the patient became more accessible in dentistry. The segmentation of the airway can be done manually or automatically. Manual segmentation seems to be the most accurate method and allows for the most operator control (*El & Palomo, 2010*). Manual segmentation needs the operator to delineate the airway slice by slice and render the data into a 3D volume for analysis. *Schendel & Hatcher (2010)* have shown that the measurement of the 3D airway from CBCT data using a semi-assisted software program is accurate, reliable, and fast. While automatic segmentation can be done by differentiating structures with different density values as done by *Shi, Scarfe & Farman (2006)* which applied a simple grayscale thresholding based method to segment and measure the upper airway using CBCT.

Accuracy and reliability of airway measurements for volume and minimum area in CBCT images have been tested. *Lenza et al. (2010)* had compared the linear, area, and volumetric measurements by two examiners and found no significant differences. *Aboudara et al. (2009)* did a study to compare the nasopharyngeal airway size between a lateral head film and a CBCT scan in adolescent subjects and found that there is a significant positive relationship between nasopharyngeal airway size on a head film and its true volumetric size from a CBCT scan. *Ghoneima & Kula (2013)* had investigated the accuracy of CBCT airway measurements by scanning the actual volume of an airway model. The results of their study showed that the CBCT digital measurements of the airway volume and the minimum area of the airway are reliable and accurate.

Automatic segmentation of the airway is significantly faster and more practical than manual segmentation and had been found that it was reliable and accurate, but the reliability and reproducibility of the method with commercially available programs were less tested. The aim of this current study was to compare and to test the reliability of the

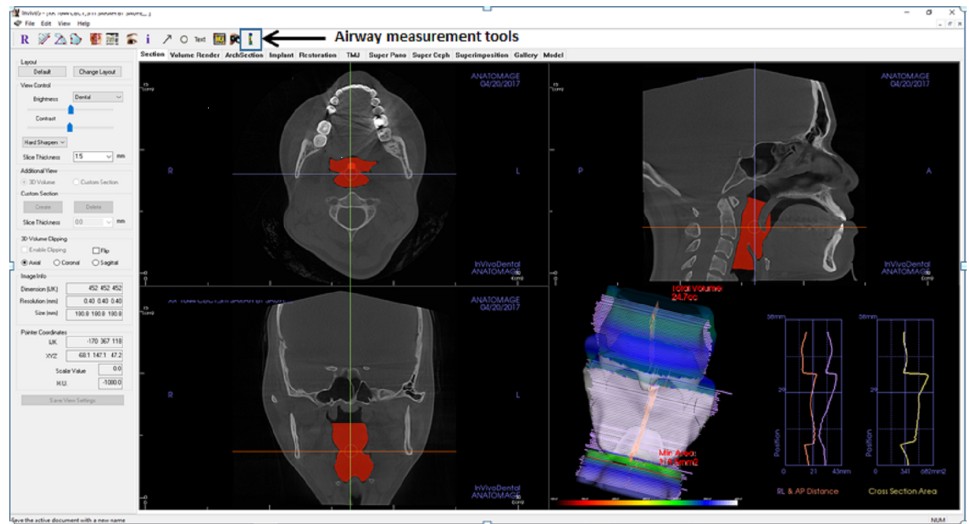

**Figure 1** **Display of Invivo5 software for airway analysis in 'section' menu.** The airway segmenting tools are shown by the arrow.

measurement of upper airway volume and minimum area using airway analysis function in two software programs: Invivo5 (Anatomage) and Romexis (version 3.8.2.R; Planmeca).

# MATERIALS & METHODS

This retrospective study was done at School of Dental Sciences, University Sains Malaysia, Health Campus, Kubang Kerian Kelantan. The sample size was calculated using a G Power calculator with $\alpha$ (probability error) of 0.05, 80% power and effect size of 0.7. From calculation, 11 samples would be sufficient. Eleven CBCT scans data were selected from the dental clinic database system, School of Dental Sciences. The CBCT scan data with the defined airway was not clear or the airway not fully contained in the image or the image containing artifacts was excluded. The entire CBCT scan data was obtained from Planmeca Promax 3D Mid (*Planmeca*, Helsinki, Finland) with 90 kV, 8 mA and 13.822 s technical factor. The scans were done using a field of view (FOV) of 160 mm, 400 $\mu$m voxel size and $454 \times 454 \times 436$ mm$^3$ image size. All the 11 CBCT scan images were analysed with airway analysis function using two software programs: Invivo5 (Anatomage) and Romexis (version 3.8.2.R Planmeca). The images were analysed by an examiner with more than three years of experience using this software.

In Invivo5, the airway was measured using the airway segmenting tool as in Fig. 1. Then the line was drawn in the middle of the airway space starting from the paranasal sinuses (PNS) level down to the middle of 4th cervical vertebra level in the sagittal view. After the line is drawn, the software will automatically detect the airway within the soft tissue based on the gray values. Once the airway has been defined and the boundaries are well established, the volume of the airway and the minimum area are automatically generated. The setting for airway analysis function can be found in 'volume render' menus as in Fig. 2.

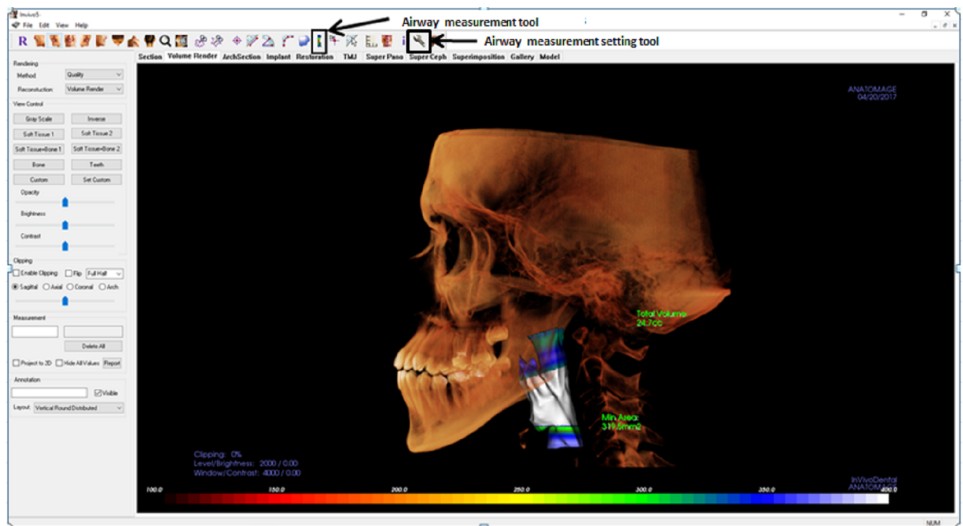

**Figure 2  Display of Invivo5 software for airway analysis in 'volume render' menu.** The airway segmenting tools are shown by the arrow.

In Romexis version 3.8.2.R software, the airway was measured using the region growing feature (as in Fig. 3). First, a cube was drawn at the area of an airway in a sagittal grayscale view using 'to draw a cube' button. The superior and inferior limit of the cube was at the PNS level and middle of the 4th cervical vertebra. The anterior and posterior limit of the cube was created by certifying that the airway boundaries were included. Then the '3D region growing' button was used to set the parameter to be used. In '3D region growing' window, the 'pre-set' box was set as 'air cavity', the threshold was set at 300, ticked at 'coloured by areas'. Next step was 'select the seed point', this step was needed to allow Romexis to know what type of density to be measured. Click on a space in the airway. Romexis then rendered up the airway and displayed the air volume and the area of the airway. However, in this software, the minimum area is not automatically displayed on sagittal view. Instead, the minimum area was searched by scrolling the axial view.

The measurement was repeated after one week. After all the measurement data was obtained, the data were analyzed using IBM SPSS software (version 23) with $t$-test to compare the measurement between software and ICC intrarater reliability test to assess the consistency of measurements made by both software in measuring the same quantity. The confidence interval was set at 95%. For intrarater reliability test, the 'model' used was 'One-Way Random'. Bland & Altman plot was then plotted to visualize the consistency between measurements.

## RESULTS

Table 1 shows the mean, standard deviation (sd) and the output from $t$-test analysis for two software. From the table, the mean airway volume and mean minimum area measurement from Romexis software are higher compared to the Invivo5 software. However, the standard deviations from Romexis measurements are lower than the Invivo5 software. The data also

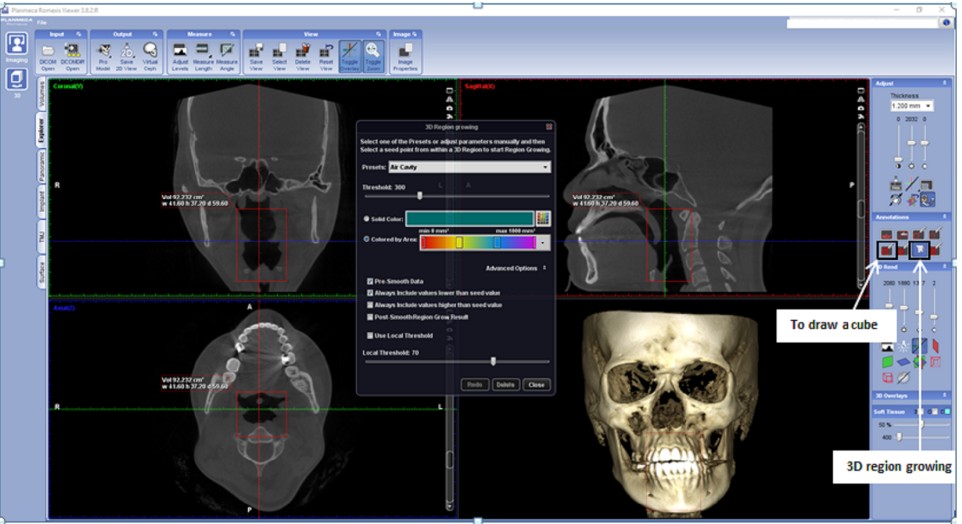

**Figure 3** Display of Romexis (version 3.8.2.R) software for airway analysis using region growing tools.
The button of 'to draw a cube' and '3D region growing' are shown by the arrow.

**Table 1** *T*-test for airway volume and minimum area.

| Quantity | Method | Mean | Std. deviation | Sig. (2-tailed) |
|---|---|---|---|---|
| Volume, cm$^3$ | Invivo5 | 17.83 | 9.48 | .914 |
| | Romexis | 18.26 | 8.86 | |
| min.area, mm$^2$ | Invivo5 | 156.97 | 89.44 | .914 |
| | Romexis | 161.00 | 83.88 | |

shows that the *p*-value (for volume and minimum area) was more than 0.05; therefore, it can be concluded that the mean reading of volume and minimum area is not significantly different between Invivo 5 and Romexis.

Table 2 shows the mean, standard deviation and output from intrarater reliability test. From the results obtained, it shows that there was evidence for the repeatability of measurements between two occasions for the software. A copy of the Bland and Altman plot for these data were shown in Figs. 4 and 5, which shows good agreement for most cases. For volume measurement, seven were nearer to zero with no outlier, and eight were nearer to zero with one outlier for Invivo5 and Romexis (Figs. 4A and 4B). For measurement of minimum area, ten were nearer to zero with one outlier and 7 were nearer to zero with one outlier for Invivo 5 and Romexis (Figs. 5A and 5B).

## DISCUSSION

There are currently more than fifteen third-party DICOM viewers mainly for orthodontics, implantology, and oral and maxillofacial surgery was available commercially. Although the reliability, repeatability and accuracy of CBCT machines have been evaluated, testing
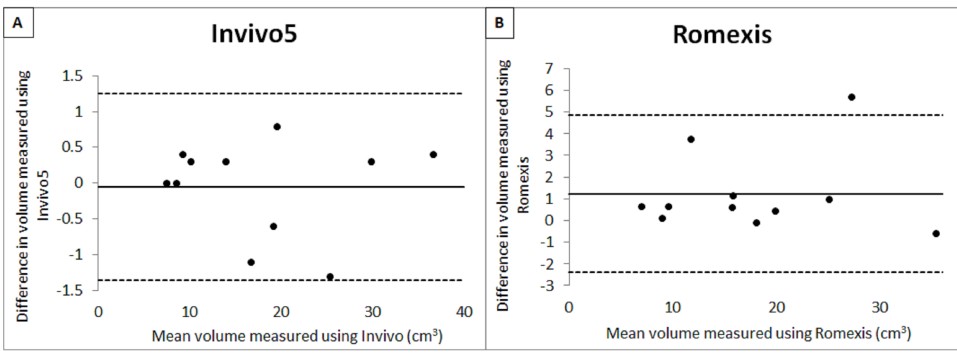

**Figure 4** Bland & Altman plot of 1 st and 2 nd measurement of volume.

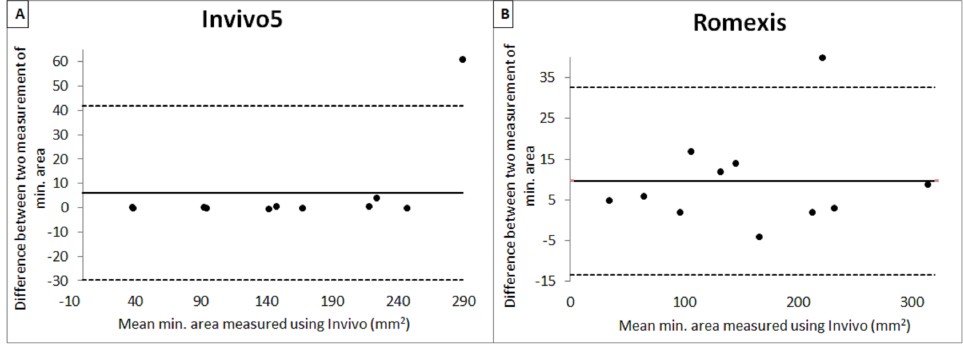

**Figure 5** Bland & Altman plot of 1 st and 2 nd measurement of min. area.

**Table 2 Intrarater reliability test (ICC) for airway volume and minimum area.**

|  | Mean | Std. deviation | Intraclass correlation (r) | Lower bound | Upper bound |
|---|---|---|---|---|---|
| Invivo5 volume, cm³ |  |  |  |  |  |
| 1st measurement | 17.83 | 9.48 | 0.998 | 0.992 | 0.999 |
| 2nd measurement | 17.87 | 9.52 |  |  |  |
| Romexis volume, cm³ |  |  |  |  |  |
| 1st measurement | 18.26 | 8.86 | 0.970 | 0.899 | 0.992 |
| 2nd measurement | 17.037 | 8.71 |  |  |  |
| Invivo5 min. area, mm² |  |  |  |  |  |
| 1st measurement | 156.97 | 89.44 | 0.976 | 0.918 | 0.993 |
| 2nd measurement | 150.91 | 79.33 |  |  |  |
| Romexis min. area, mm² |  |  |  |  |  |
| 1st measurement | 161.00 | 83.88 | 0.984 | 0.945 | 0.996 |
| 2nd measurement | 151.36 | 81.83 |  |  |  |

the reliability of CBCT-related software has not gone further as they differ in terms of the statistical test used.

In this study, two commercially available CBCT software programs that use automatic segmentation to calculate airway volumes were tested. From the *t*-test analysis, the *p*-value is equal to 0.914 for both quantity measured. This means that there is no significant difference between the two software for the airway volume and minimum area. While for ICC test, the intrarater value is more than 0.90 indicating excellent agreement. According to *Koo & Li (2016)*, the ICC value of 0.75 and above is considered as excellent. So, the correlation values obtained from this study indicate that they are reproducible. The results obtained are supported by other studies (*El & Palomo, 2010*; *Ghoneima & Kula, 2013*; *Lenza et al., 2010*; *Feng et al., 2015*). *Petdachai & Chuenchompoonut (2017)* had used Romexis software to find the correlation between 3D airway and 2D. They found that the correlation value between the area in 2D and volume in 3D are very high correlation. While for Invivo software, *Kim et al. (2010)* had used this software to measure pharyngeal airway volumes in healthy children with retrognathic mandibles and those with normal craniofacial growth.

The measurement from this software differs slightly due to the fact that these software programs did not use the same methods for calculation of the airway volume and the minimum area. In Invivo5, the segmentation of the airway was based on the point the user click on the airway space and the upper and lower level are follows the shape of the airway. However, in Romexis, the segmentation was done base on the region growing in a cube, thus the upper and lower level does not follow the shape of the airway. This gives a slightly a variation of measurement for both software. The Invivo5 software allows more control where the user can "sculpt out" the desired airway volume from the rest of the 3D structures. The user also can adjust the brightness and opacity values, clean out the unwanted voxels before calculating the final airway volume. The software also lets the user to change the threshold values to obtain a solid airway volume. This also might be the reason to why the measurement of volume using Invivo5 software is more variable than Romexis software.

For automatic segmentation, volume measurements should be done with proper technique and diligence. This is because the measurement changes depending on the image threshold chosen. This is proved by *El & Palomo (2010)*. The proper technique also important as the different position will significantly increase or decrease the measurement (*Camacho, Capasso & Schendel, 2014*). A study had proved that the CBCT-based 3D analysis gives a better picture of the anatomical characteristics of the upper airways and therefore can lead to an improvement of the diagnosis (*Lenza et al., 2010*). The automatic segmentation of the airway imaged using CBCT is feasible and this method can be used to evaluate airway cross-section and volume comparable to measurements extracted using manual segmentation (*Shi, Scarfe & Farman, 2006*). *Ghoneima & Kula (2013)* had suggested that the three-dimensional CBCT digital measurements of the airway volume and the most constricted area of the airway are reliable and accurate. The use of CBCT imaging for the assessment of the airway can provide clinically useful information in orthodontics and for assessing the airway after surgery. This is proved by *Alsufyani et al. (2017)* where they concluded that the use of point-based analysis (from

3D CBCT) measures is better explained the changes in clinical symptoms compared to conventional measures. *Yamashina et al. (2008)* had evaluated the reliability of CBCT values and dimensional measurements of oropharyngeal air spaces as compared to multidetector CT on the phantom and clinical patient. They found that the measurement of air spaces with CBCT was quite accurate.

The Bland & Altman plot created to compare the two measurements that each provides some errors in their measure. The plot also allows the identification of any systematic difference between the measurements or possible outliers. The dotted horizontal lines represent the 95% confidence limits (limits of agreement). Thus, if the differences between methods were distributed normally, 95% of the differences from the bias in the sample are expected to be between the upper and lower limit of agreement. As the confidence limits are not exceeded, it can be concluded that the repeatability of the method is acceptable and the two methods are considered to be in agreement and may be used interchangeably.

## CONCLUSIONS

From this study, both Romexis 3.8.2.R and Invivo5 are shown not to give significantly different readings and are reproducible in their volume and minimum area measurements. If available, both software programs can be used interchangeably.

## ACKNOWLEDGEMENTS

Thank you to all staff involved from dental clinic for their cooperation and support.

### Funding
This work was supported by USM Research University Team grant number 1001/PPSG/852004. The funders had no role in study design, data collection and analysis, decision to publish, or preparation of the manuscript.

### Grant Disclosures
The following grant information was disclosed by the authors:
USM Research University Team: 1001/PPSG/852004.

### Competing Interests
The authors declare there are no competing interests.

### Author Contributions
- Noorshaida Kamaruddin conceived and designed the experiments, performed the experiments, analyzed the data, prepared figures and/or tables, authored or reviewed drafts of the paper.
- Firdaus Daud conceived and designed the experiments, contributed reagents/materials/analysis tools, authored or reviewed drafts of the paper.
- Asilah Yusof and Mohd Ezane Aziz verify the measurement and check the manuscript.
- Zainul A Rajion authored or reviewed drafts of the paper, approved the final draft.

## Data Availability

The raw data are available in the Supplemental Files.

## Supplemental Information

Supplemental information for this article can be found online at http://dx.doi.org/10.7717/peerj.6319#supplemental-information.

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
