# Peer review of "Comparison of automatic airway analysis function of Invivo5 and Romexis software"

_PeerJ, doi:10.7717/peerj.6319_

## Round 0.1 · original submission · Major Revisions

An important issue that is raised by both reviewers is how the boundaries of the airway space were defined. please describe this very clearly in your revision as this information is essential.

Besides the other methodological comments of the reviewers that should be addressed I must stress that you should have your paper copy-edited by a professional academic editing service as language needs improvement.

Please also have a look at your references as there are inconsistencies and mistakes.

Reviewer 1 ·

Basic reporting

The manuscript is well written
The references are current but could be more explored (there are a lot of studies exploring this issue)
The structure of the article is ok
The keywords should represent the general aim of the study and not specify the software used.
It's important to comment on the introduction the differences between the studies that have gold standard and the studies that don’t have.

Experimental design

Did the authors calculate the sample size? Why did they use 11 CBCT scans?
How the authors define the boundaries of the upper airway space? This is essential information that is not clear in the text.
About the examiner. Its important cite the experience with this software….

Validity of the findings

Its important compare software but I think that specific points that could be decisive were not discussed as the CBCT protocols, the boundaries of the upper airway, the soft tissues during the acquisition.These issues can influence the airway measurements. Maybe compare software should be a minor finding.

I think that this manuscript doesn’t add any new and important information; it's natural to believe that if you have an automatic segmentation and you repeat the measurements you will have replicated sizes as well. Maybe they have similar results but don’t have a gold standard, so the measures are actual? At least this should be discussed.

Bland Altmann is currently used to compare a standard method and a new one, and it's a good way to visualize if exist differences between the two techniques.

·

Basic reporting

A. The English language used has many grammatical mistakes and is often unclear. The authors use a mix of the present and the past tense while it is custom to report finding in the past tense. The article needs major professional English language editing before it could be suitable for publication. A few examples where the English language could be improved/corrected are lines 24, 31, 41, 51-53, 70, 137, 144.

B. The references are NOT formatted according to the journal's style and contain many mistakes. The journal style has the volume only while they have stated the volume and the issue in many references. Authors names should be last name then first name initials, many references have first name and then last and therefore are not properly alphabetically ordered. Eg the first reference line 187 “ Oral Radiology, 0(0), 0.” The volume is definitely not zero, it is vol 33 and the pages are not stated.

C. The word “reference” is present a couple of times in the text prior to the in text citation ( lines 136, 162)

D. Naming of the minimum cross-sectional area (MCA) is not consistent in the study, they start by MCA and then they just use minimum area in the rest of the text. Please standardize how you refer to this measurement throughout the text.

E. In the results section the authors report in details numbers/figures that are already shown in table 2 so there is no need to repeat them in the text.

Experimental design

A. I would like to ask the authors to specify how they defined/standardized the upper and lower boundaries of the airway? Whether they included the nasal airway in their measurements or not? This information is important to ensure the repeatability/ Reproducibility of the measurement especially when drawing the “cube” in the Romexis software.

B. The Methods described are lacking the machine settings (scanning parameters) used when acquiring the CBCT scans. The authors are using software programs that depend on a set grey value to automatically segment the airways. Since the gray value of a pixel depends not only on the tissue contrast but also on many other factors, such as the type of CBCT machine used and the scanning parameters. I would like to ask the authors to report the scanning parameters used such as FOV, voxel size, etc. to ensure sufficient details and information to replicate the study.

C. In line 83-84 the authors state “ After the line is drawn, the software will automatically detect the airway within the soft tissue based on the Hounsfield Unit.” Since the images used in the study are CBCT images, then the software would be detecting the soft tissue based on the Grey Values not on HU. Please correct this in the text.

Validity of the findings

No comment

---

## Round 0.2 · Major Revisions

Thank you for your resubmission and the revisions you made. The paper had been re-reviewed. The referee still has comments and I agree with all of them. Please address the comments.

As I stated before, please have your paper copy-edited by a professional academic editing service as we cannot publish it as it is now. I would suggest to contact www.sfedit.net in San Francisco (US) as I have good experiences with their work. However, you are completely free to use any professional service you would like.

Furthermore, Table 2 and 3 give measurements for airway volume and minimum area, but the unit of measurements is not given. Is this cubic and square mms? Please indicate. Numbers in 4 decimals is also overdone for this type of measurements, please round numbers to 2 decimals.

Figure 4 and 5 (Bland Altman plots): I miss a scale on the X-axis for the mean values.

·

Basic reporting

The authors have added the needed information about their research methodology, corrected their references as requested but from the English language point of view the article, in it's present state, is NOT suitable for publication. Many sentences start in the present and end in the past, many don't even have a verb and are difficult to understand. To name a few examples " line 179 : " this gives a slightly measurement for both software", line 155: There are currently more than fifteen third party DICOM viewers .... was available commercially." The authors are again advised to have their article edited by a professional academic editing service

Experimental design

The authors described in their materials and methods section how they defined the airway boundries in the Invivo5 software. However, they did not describe how they reproduced the same boundries when using the Romexis software.

Validity of the findings

no comment

---

## Round 0.3 · accepted · Accept

I have checked your revisions and you addressesd the comments of the reviewers in this second revision.